# NEURAL RANDOM PROJECTIONS FOR LANGUAGE MODELLING

## ABSTRACT

Neural network-based language models deal with data sparsity problems by mapping the large discrete space of words into a smaller continuous space of real-valued vectors. By learning distributed vector representations for words, each training sample informs the neural network model about a combinatorial number of other patterns. In this paper, we exploit the sparsity in natural language even further by encoding each unique input word using a fixed sparse random representation. These sparse codes are then projected onto a smaller embedding space which allows for the encoding of word occurrences from a possibly unknown vocabulary, along with the creation of more compact language models using a reduced number of parameters. We investigate the properties of our encoding mechanism empirically, by evaluating its performance on the widely used Penn Treebank corpus. We show that guaranteeing approximately equidistant vector representations for unique discrete inputs is enough to provide the neural network model with enough information to learn –and make use– of distributed representations for these inputs.

## 1 INTRODUCTION

Representation learning is integral to a wide range of tasks from Language Modelling (LM) to speech recognition, or machine translation. Better language models frequently lead to improved performance on underlying downstream tasks, which makes LM valuable in itself. The goal of LM is to model the joint probability distribution of words in a text corpus. Models are trained by maximising the probability of a word given all the previous words in the training dataset:

$$P(w_1, \ldots, w_N) = \prod_{i=1}^{N} P(w_i|h_i),$$

where $w_i$ is the current word and $c_i = (w_1, \cdots, w_{i-1})$ is the current word *history* or *context*. A Markov assumption is usually used as approximation: instead of using the full word history as context, the probability of observing the $i^{th}$ word $w_i$ is approximated by the probability of observing it in a shortened context of $n-1$ preceding words. To train a language model under this assumption, a *sliding window* $(w_i, c_i)$ of size $n$ is applied sequentially across the textual data. Each window is also commonly referred to as *n-gram*.

Neural Network Language Models (NNLM) (Bengio et al., 2003) deal with both data sparseness and generalisation problems, by using neural networks to model the probability distribution for a word given its context, as a smooth function of learned real-valued vector representations for each word. Neural networks thus learn a conditional probability function $P(w_i|c_i; \theta_F, \theta_L)$, where $\theta_F$ is a set of parameters representing the vector representations for each word and $\theta_L$ parameterises the log-probabilities of each word based on those learned features. A softmax function is then applied to these log-probabilities to produce a categorical probability distribution over the next word given its context. The resulting probability estimates are smooth functions of the continuous word vector representations, and so, a small change in such vector representations results in a small change in the probability estimation.

Energy-based models provide a different perspective on statistical language modelling. Energy-based models such as (Mnih & Hinton, 2007) use neural networks to capture dependencies between

variables by associating a scalar energy score to each output configuration. The key difference from the previously referred approach to NNLM, is that instead of trying to probability of a target word directly, the models predict the distributed representation for the target word (embedding) and attribute an energy score to the output configuration of the model based on how close the prediction is from the actual representation for the target word.

In this paper, we describe an encoding mechanism that can be used with neural network probabilistic language models to encode each word without using a predetermined vocabulary of size $|V|$. Each unique word is encoded using a fixed sparse $k - dimensional$ random vector with $k \ll |V|$. These random features are designed so that the inner products of the original feature space is approximately preserved (Achlioptas, 2003). Critically, we show that we can achieve results comparable to the typical NNLM, without learning a unique embedding vector for each word.

## 2 BACKGROUND AND RELATED WORK

One cornerstone of neural-network-based models is the notion of distributed representation. In a distributed representation, each entity is represented as a combination of multiple factors. Learning distributed representations of concepts as patterns of activity in a neural network has been object of study in (Hinton, 1986). The mapping between unique concept ids and respective vector representations has been later referred to as *embeddings*. The idea of using embeddings in language modelling is explored in the early work or Bengio et al. (Bengio et al., 2003). This and similar approaches, encode input words using a $1$-of-$V$ encoding scheme, where each word is represented by its unique index. The size of the embedding space is thus proportional to the size of the vocabulary, making neural language modelling computationally taxing for large vocabularies.

One straightforward method to deal with large vocabularies in LM, is to consider character or subword-level modelling instead of words as the input units such as the work in (Kim et al., 2016). Another motivation behind this approach, is the fact that word-based models often fail to capture regularities in many inflectional and agglutinative languages. In this paper we focus on word-based modelling, but the proposed encoding mechanism can be incorporated in models that use for example morphemes instead of words as inputs.

Another difficulty with building neural language models lies in the fact that outputting probability distributions require explicit normalisation: we need to consider the output probabilities for all words in the vocabulary to compute log-likelihood gradients. Solutions to this problem include structuring the vocabulary into a tree, speeding-up word probability computations (Morin & Bengio, 2005), or approximating the log-likelihood gradient using approaches like Noise Constrastive Estimation (NCE) (Mnih & Teh, 2012). In order to analyse the predictive capability of our proposed models, we will use fully normalised models. This means we are still dependent on a known vocabulary to train the model. The normalisation requirements can be alleviated using techniques such as the ones previously mentioned, but, this will be the subject for future explorations.

Exploiting low-dimensional representations for structures in high-dimensional problems has become a highly active area of research in machine learning, signal processing, and statistics. In summary, the goal is to use a low-dimensional model of relevant data in order to achieve, better prediction, compression, or estimation compared to more complex "black-box" approaches to deal with high-dimensional spaces (Hegde et al., 2016). In particular, we are interested in the compression aspect of these approaches. Usually, exploiting low-dimensional structure comes at a cost: incorporating structural constraints into a statistical estimation procedure often results in a challenging algorithmic problem but neural network models have been notoriously successful at dealing with these kind of constraints.

The use of random projections as a dimensionality reduction tool has been extensively studied before, especially in the context of approximate matrix factorisation –the reader can refer to (Halko et al., 2011) for a more in depth review of this technique. The basic idea of random projections as dimensionality reduction technique comes from the work of *Johnson and Lindenstrauss*, in which it was shown that the pairwise distances among a collection of $N$ points in an Euclidean space are approximately maintained when the points are mapped randomly to an Euclidean space of dimension $O(\log N)$. In other words, random embeddings preserve Euclidean geometry (Johnson & Lindenstrauss, 1984).

Random projections have also been used in multiple machine learning problems as a way to speed-up computationally expensive methods. Examples include the use of random mappings to accelerate the training of kernel machines (Rahimi & Recht, 2008). In (Papadimitriou et al., 2000), random projections are used as a first step for Latent Semantic Analysis (LSA), which is essentially a matrix factorisation of word co-occurrence counts. A similar approach is followed in (Kaski, 1998): a random mapping projection is applied to document vectors to reduce their dimensionality before an unsupervised clustering algorithm is applied these vectors. The random mappings we use in this paper, are motivated by the work in (Achlioptas, 2003), which showed that a random projection matrix can be constructed incrementally using a simple sparse ternary distribution.

In this paper, we focus on exploring simple feedforward neural network architectures instead of more complex and large recurrent neural network models such as (Jozefowicz et al., 2016). This will allow us to iterate quickly on experiments, and better understanding the effects of random projections on network performance. As a starting point to build a baseline model, we chose an architecture similar to (Bengio et al., 2003) along with the the energy-based formulation in (Mnih & Hinton, 2007). We chose this approach because it gives us one of the simplest forms of a neural language model. Furthermore, in the future, the implementation of log-likelihood approximations such as NCE becomes straightforward under this formulation.

## 3 MODEL DESCRIPTION

This section describes our baseline neural language model, along with our alternative formulation using a random projection encoding. The main idea behind the baseline model is to use a neural network to learn lower-dimensional representations for words, while learning a discrete probability distribution of words, similarly to (Bengio et al., 2003; Mnih & Hinton, 2007). While the baseline model uses a 1-of-$V$ vector to represent each input word, and a $|V| \times m$ embedding projection layer; the neural random projection model encodes each word using a random sparse $k - dimensional$ vector and $k \times m$ embedding layer.

### 3.1 RANDOM INPUT ENCODING

The distributed representations or *embeddings* in our model are $k \times m$ subspace, with $k << |V|$. A representation for each word as learned by the neural network is the linear combination of $(s \times m)$-dimensional basis where $s$ is the number of non-zero entries for each vector. Each unique word is assigned a randomly generated sparse ternary vector we refer to as *random index vector*. Random indices are sampled from the following distribution with mean 0 and variance 1:

$$r_i = \begin{cases} +1 & \text{with probability } \alpha/2 \\ 0 & \text{with probability } 1 - \alpha \\ -1 & \text{with probability } \alpha/2 \end{cases} \qquad (1)$$

where $\alpha$ is the proportion of non-zero entries in the random index vector. While similar to the distribution proposed in (Achlioptas, 2003; Kanerva et al., 2000), we use very sparse random vectors and no scaling factor. Since the input space is an orthogonal unit matrix, the inner product between any two random vectors $(r_i, r_j)$ is expected to be concentrated at 0. The motivation for this is to make any discrete input indistinguishable from any other input in terms of distance between them. We will use $s$ to refer to number of non-zero entries for randomly generated index vectors. This means that for $s = 2$ we have exactly one entry with value 1, one with value $-1$ and all the rest have value 0. Figure 1 shows the distribution of inner products between randomly generated vectors with different numbers of non-zero entries.

### 3.2 BASELINE MODEL

In order to assess the performance of our added random projection module, we build a **baseline** model *without* random projections. The architecture for this model is based in the one in (Bengio et al., 2003) but instead of predicting the probability of each word directly we use the energy-based definition from Mnih's work (Mnih & Hinton, 2007). The baseline model learns to predict the

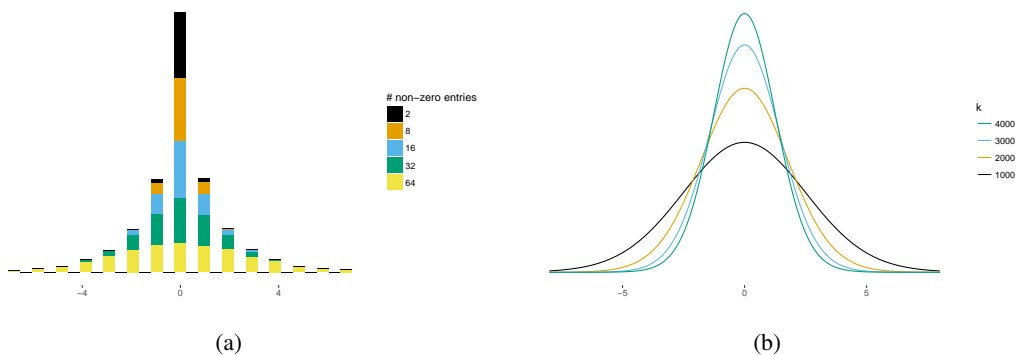

(a)            (b)

Figure 1: Inner product distribution for different number of non-zero entries $s \in \{2, 8, 16, 32, 64\}$ and $k = 1000$ (1a), and different random index dimensions $k$ aggregated by $s$ (1b).

feature vector (embedding) for a target word along with a probability distribution for all the known words conditioned on the learned feature vectors of the input context words.

The baseline language model in figure converts each word into a vector of real-valued features using a feature lookup table $F$. The goal is to learn a set of feature vectors in $F$ that are good at predicting the feature vector for the target word $w_n$. Each word in a context of size $n - 1$ is converted to $n - 1$ $m$-dimensional feature vectors that are then concatenated into a $(n - 1) \times m$ vector and passed to the next layer $h$ that applies a non-linear transformation:

$$h_j = \sigma \left( W_h f + b_h \right) \tag{2}$$

where $f$ is a vector of $n - 1$ concatenated feature vectors, $b$ is a bias for the layer $h$ and $\sigma$ is a non-linear transformation (e.g. sigmoid, hyperbolic tangent, or rectifier linear unit (ReLU)). We found ReLU units $f(x) = max(0, x)$ to perform well and at an increased training speed, so we use this transformation in our models. The result of the non-linear transformation in $h$ is then passed to a linear layer $\hat{y}$ that predicts the feature vector $\hat{f}_n$ for the target word $w_n$ as follows:

$$\hat{y}_n = W_{\hat{y}} h + b_{\hat{y}} \tag{3}$$

The energy function $E$ is defined as the dot product between the predicted feature vector $\hat{f}_n$, and the actual current feature vector for the target word $f_n$:

$$E_n = \hat{f}_n^T f_n \tag{4}$$

To obtain a word probability distribution in the network output, the predicted feature vector is compared to the feature vectors of all the words in the known dictionary by computing the energy value (equation 4) for each word. This results in a set of similarity scores which are exponentiated and normalised to obtain the predicted distribution for the next word:

$$P(w_n | w_1 : n - 1) = \frac{\exp(E_n)}{\sum_j^{|V|} \exp(E_j)} \tag{5}$$

The described baseline model can be though of as a typical feedforward model as defined in (Bengio et al., 2003) with an added linear layer $\hat{y}$ with dimension $m$. Furthermore, instead of using two different sets of embeddings to encode context words and compute the word probability distributions, we learn a single set of feature vectors $F$ that are shared between input and output layers to compute the energy values.

### 3.3 MODEL WITH RANDOM PROJECTIONS

We incorporate our random projection encoding into the previously defined energy-based neural network language model following the formulation in (Mnih & Hinton, 2007). We chose this formulation because it allows us to use maximum-likelihood training and evaluate our contribution in terms of *perplexity*. This makes it easier to compare it to other approaches that use the same intrinsic measure of model performance. It is well known that computing the normalisation constant (the denominator in equation 5) is expensive, but we are working with relatively simple models, so we compute this anyway.

Having defined the baseline model, we now extend it with a random projection lookup layer. The overview of the resulting architecture can be seen in figure 2.

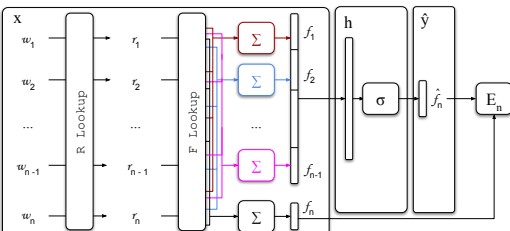

Figure 2: Neural random projection feedforward energy model.

The NRP model uses random index lookup table that generates a new random index for each new word that it encounters. Tho get the feature representation for a given word $w_i$ we multiply its $k-$ dimensional random index by the $k \times m$ feature matrix $F$. The resulting vector representation for a word $w_i$ is given by the sum of all the feature vectors extracted for each non-zero entry in the random index. Only the rows corresponding to the non-zero entries are modified during training for each input sample. This adds to the computational complexity of the first layer since we are no longer selecting a single feature vector for each word, but it is still quite efficient since the random indices are sparse. All models sare trained using a maximum likelihood criterion:

$$\mathcal{L} = \frac{1}{T} \sum_t \log P(w_t, w_t - 1, \ldots, w_t - n + 1; \theta) \tag{6}$$

Where $\theta$ is the set of parameters learned by the neural network in order to maximise a corpus likelihood.

## 4 EXPERIMENTAL SETUP

We evaluate our models using a subset of the *Penn Treebank* corpus, more specifically, the portion containing articles from the Wall Street Journal. We use a commonly used set-up for this corpus, with pre-processing as in (Mikolov et al., 2011), resulting in a corpus with approximately 1 million tokens (words) and a vocabulary of size $|V| = 10K$ unique words The dataset is divided as follows: sections 0-20 were used as training data (930K tokens), sections 21-22 as validation data (74K tokens) and 23-24 as test data (82K tokens). All words outside the 10K vocabulary were mapped to a single token $< unk >$. We train the models by dividing the PTB corpus into *n-gram* samples (sliding windows) of size $n$. All the models reported use *5-grams*. To speed up training and evaluation, we batch the samples in mini-batches of size 128 and the training set is randomised *prior* to batching. Furthermore, in these experiments the n-grams do not cross sentence boundaries.

We train the models by using the *cross entropy* loss function and evaluate and compare the models based on their perplexity score:

$$PPL = \exp\left(-\frac{1}{N} \sum_{w_{1:n}} log P(w_n | w_{1:n-1})\right) \tag{7}$$

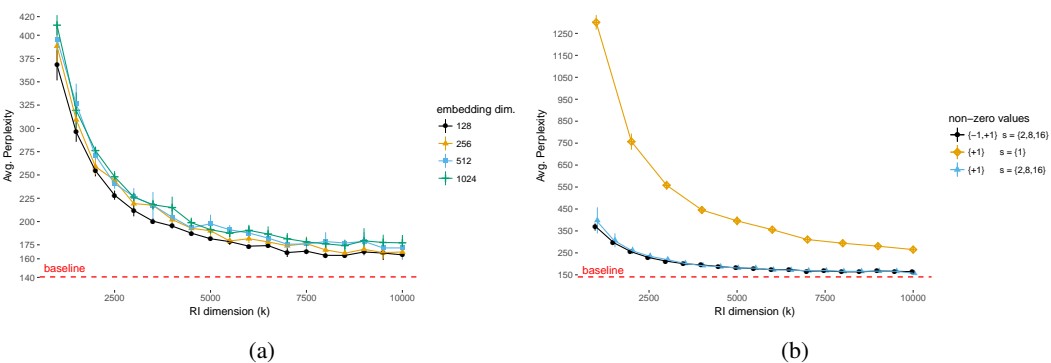

Figure 3: Average *test* perplexity for multiple embedding sizes $m \in \{128, 256, 512, 1024\}$ and number of hidden units set to $h = 256$: (3a) results aggregated by number of non-zero entries $s \in \{2, 8, 16\}$; (3b.) comparison between ternary random index vectors $\{+1, 0, 1\}$, binary random index vectors, and random index vectors with single entry $\{+1\}$ using an embedding size of $m = 128$.

where the sum is performed over all the *n-gram windows* of size $n$. We train our models using Stochastic Gradient Descent (SGD) without momentum. In early experiments, we found that in our setting, SGD produced overall better models and generalised better than other adaptive optimisation procedures. We use a step-wise learning rate annealing: the *learning rate* is kept fixed during a single epoch, and multiplied by $0.5$ every time the validation perplexity increases at the end of a training epoch. If the validation perplexity does not improve, we keep repeating the process until 3 epochs (*patience* parameter) have passed without improvement (early stop). We consider that the model **converges** when it stops improving and the *patience* parameter reaches $0$

We used Rectifying Linear Units (ReLU) units (He et al., 2015) as non-linear activations for the hidden layers. Notwithstanding their good performance, ReLU units caused gradients *explode* easily during training. To mitigate this, we added *local* gradient norm clipping to the value $1.0$.

*Dropout regularisation* (Srivastava et al., 2014) is applied to all weights including the feature lookup (embeddings). In early experiments, we used a small dropout probability of $0.05$ following the methodology in (Pham et al., 2016). We later found higher dropout probability values to give us better performance, when used in conjunction with larger embedding and hidden layer sizes.

All network weights are initialised randomly on a range $[-0.01, 0.01]$ in a random uniform fashion. Bias values are initialised with $0$. We found it beneficial to initialise the non-linear layers with the procedure described in (He et al., 2015).

## 5 EXPERIMENTAL RESULTS

Initial experiments explored the effects of different random index (RI) configurations in the model perplexity scores (less is better). In these first experiments, we use a single hidden layer of size $h = 256$, while the dropout probability is fixed to $0.05$. We span three different parameters: embedding size $m \in \{128, 256, 512, 1024\}$, RI dimension $k \in [1000, 10000]$, and the number of non-zero entries in the random projections $s \in \{2, 8, 16\}$.

Figure 3a shows the results for the average test perplexity scores aggregated by the number of non-zero entries in the RIs. The perplexity score follows an exponential decay as the RI dimension $k$ increases. From $k >= 5000$, the perplexity values converge to an approximate range $PPL \approx [170, 180]$. Subsequently, we found out that a more aggressive regularisation improves both the NRP and baseline models. Curiously, but perhaps not surprisingly, the relationship between random index dimension $k$ and perplexity scores seems to resemble the relationship between random projection dimension dimension and distance error bounds, for the random projection method described in (Johnson & Lindenstrauss, 1984).

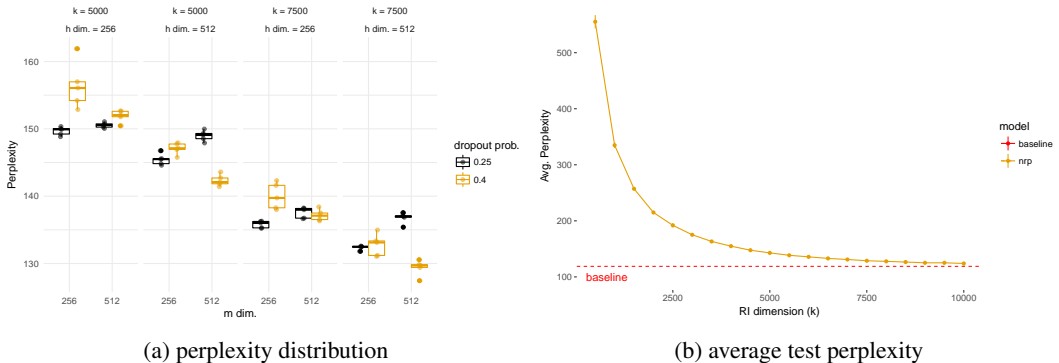

(a) perplexity distribution                    (b) average test perplexity

Figure 4: (4a) Distribution of test perplexities for NRP model with $h \in \{256, 512\}$ and dropout $p \in \{0.25, 0.4\}$; (4b) average *test* perplexity for best NRP and baseline configurations (table 1) with $m = h = 512$ and *dropout* $p = 0.4$ for multiple values of $k$.

In another experiment, we looked at whether or not there was a difference in performance between NRP models using ternary random indices (with values $\in \{-1, 0, 1\}$), and models using binary random indices (with values $\in \{0, 1\}$). We also wanted to find if redundancy (having more non-zero entries) is key to the NRP model performance. To answer this second question, we tested models that used random indices with a single non-zero value. In this setting, different words can share the same feature vector. This increased collision chance is problematic as we can see in figure 3b, where we shows the results for this experiment. As expected, having a single non-zero positive entry $s = 1$ yields worse results than any other configuration because the any collision between random indices makes so that the network cannot distinguish features of the colliding words.

Having just two non-zero entries $s >= 2$ is enough to improve the model performance in terms of perplexity. It seems that using binary and ternary random indices yields similar results, with binary indices displaying an increased variance for lower values of random index dimension $k$. We think this increase in variance is due to the fact that lower-dimensional binary random indices have a higher probability of collisions and thus disambiguation becomes harder to perform. As for why the results are similar, we should note that the near-orthogonality properties of random indices are still maintained whether one uses binary or ternary representations. Moreover, the feature layer (or embeddings) is itself initialised with symmetric random weights, having equally distributed positive and negative values, so the symmetry of the input space should not impact on the model performance too much for larger dimensions.

## 5.1 MODELS WITH BETTER REGULARISATION

After additional testing and fine-tuning, we found models with a better performance using a higher dropout probability along with larger hidden layer and embedding sizes (see figure 4a and table 1). We should mention that even with half the embedding parameters (using random indices of dimension $k = 5k$ to encode a vocabulary of $10k$ words), we get reasonably good models with test perplexities of 145 to 150 using only, 1.6 to 1.9 million parameters. We can compare this with the test perplexity scores of 109 to 149, obtained on the same dataset in (Pham et al., 2016) with similar models and 4.5 million parameters. The comparison is not really fair in this case because the authors use larger n-gram windows, but we can still see that our baseline yields reasonably good results. This makes our relatively simple language model, a good starting point to study the effect of random projections in this particular task.

Additional tests showed that we cannot go much further with dropout probabilities, so we use a value of 0.4, since it yielded the best results for both the baseline and the models using random projection encoding. One weakness in our methodology is that hyper-parameter exploration was done based on grid search and intuition about model behaviour –mostly because each model run is very expensive even for our simple architectures. In future work this methodological aspect will be improved by replacing grid search by a more appropriate method such as *Bayesian Optimisation*.

Finally, we analyse the behaviour of the NRP model when varying the random projection dimension $k$, and compare it with the best baseline found. The results can be seen in figure 4b. As with our early experiments, the perplexity decays exponentially with the increase of the random index dimension $k$. One key difference is that this decay is much more accentuated and model test perplexity converges to values closer to the baseline. In summary, random projections seem to be a viable technique to build models with comparable predictive power, with a much lower number of parameters and without the necessity of *a priori* knowledge of the dimension of the lexicon to be used. Proper regularisation and model size tuning was key to achieve good results. NRP models are also more sensitive to hyper-parameter values and layer sizes than the baseline model. This was expected, since the problem of estimating distributed representations using a compressed input is considerably harder than using dedicated feature vectors for each unique input.

Table 1: Average test perplexity and epochs for best baseline and NRP models with multiple values of *random index/input vector* dimension $k$ embedding size $m$, hidden units $h$ and dropout probability *drop*. The number of non-zero entries in the random projections is set fo 4. $\#p$ is the approximate number of trainable parameters in millions.

| | | | | | PPL | | Epoch | |
|---|---|---|---|---|---|---|---|---|
| $k$ | $m$ | $h$ | $\#p$ | *drop* | Avg. | SD | Avg. | SD |
| $7,500$ | 256 | 256 | 2.2 | .25 | 136 | 0.5 | 22 | 1.3 |
| | | | | .40 | 140 | 1.9 | 26 | 2.9 |
| | | 512 | 2.5 | .25 | 132 | 0.3 | 21 | 0.8 |
| | | | | .40 | 133 | 1.6 | 24 | 2.3 |
| | 512 | 256 | 4.4 | .25 | 137 | 0.8 | 21 | 0.8 |
| | | | | .40 | 137 | 0.8 | 24 | 1.7 |
| | | 512 | 5.1 | .25 | 137 | 0.8 | 18 | 1.8 |
| | | | | .40 | **129** | 1.2 | 23 | 1.5 |
| baseline | | | | | | | | |
| $10,000$ | 512 | 512 | 6.4 | .40 | **118** | 0.4 | 22 | 1.3 |

## 6 CONCLUSION AND FUTURE WORK

We investigated the potential of using random projections as an encoder for neural network models for language modelling. We showed that using random projections allows us to obtain perplexity scores comparable to a model that uses an 1-of-$V$ encoding, while reducing the number of necessary trainable parameters. Our goal is not to find the best possible neural language model, but to show that we can achieve viable results with random projections and a reduced embedding space. We also introduced a simple and reproducible baseline on the small, but widely used *Penn Treebank* dataset, with qualitatively comparable results with the existing literature. Our baseline model combined the energy-based principles from the work in (Mnih & Hinton, 2007) with the simple feedforward architecture proposed in (Bengio et al., 2003), using a more recent regularisation technique (dropout (Srivastava et al., 2014)) and activation units (ReLU).

Understanding how model performance is bounded by inner product distortion errors caused by random projections still warrants further investigations. Nevertheless, one exciting future direction, is to study the feasibility of using log-likelihood approximation techniques (such as Noise Constrastive Estimation (NCE)) to models using random projections, eliminating the need for a known vocabulary completely from density estimation problems. This would allow us to trade an acceptable amount of error for the capacity of training models in an incremental fashion.

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
