# OpenReview forum: "Neural Random Projections for Language Modelling"
_ICLR.cc/2019/Conference_

### Official Review · AnonReviewer2 · 2018-10-30
**Preliminary work on using random projections for word embeddings in language models**

**Rating:** 3
**Confidence:** 4

**Review:**

The main idea behind the paper is to use random projections as the initial word representations, rather than the vocab-size 1-hot representations, as is usually done in language modeling. The benefit is that the matrix which projects words into embedding space can then be much smaller, since the space of random projections can be much smaller than the vocab size. The idea is an interesting one, but this work is at too much of a preliminary stage for a top-tier conference such as ICLR. In its present state it would make for a potentially interesting paper at a targeted workshop.

More specific comments
--

The initial description of the language modeling problem assumes a particular decomposition of the joint probability, according to a particular application of the chain rule, but of course this is a modeling choice and not the only option (albeit the standard one).

The main problem with the paper is the use of simple baseline setups as the only experimental configuration:

o feedforward rather than recurrent network;
o use of the Penn Treebank dataset only;
o use of a small n for the n-grams.

All or at least some of these decisions would need to be relaxed to make a convincing paper.

The reasons for the use of the energy-based formulation are not clear to me. Is the energy-based model particularly well-suited to the random-projection setup, or are there other reasons for using it, independent of the use of random projections?

Just before equation 6 it says that the resulting vector representation is the *sum* of all the non-zero entries. But there are some minus ones in the random projection?

The PPL expression at the bottom of p.5 doesn't look right. The index over which the sum happens is n, but n is fixed? So this looks like a sum with just one component in it, namely the first n-gram.

It looks like all the results are given on the test set. Did you not do any tuning on the validation data?

The plots in figure 4 are too small. It would be useful to have a table, like the one on the last page, which clearly shows the baseline vs. the random-projection model, with some description of the results in the main body of the text.

The overall presentation could be better, and I would encourage the authors to tidy the paper up in any subsequent submission. For example, there are lots of typos such as "instead of trying to probability of a target word".

---

> ### Author Response · Authors · 2018-11-27
> **response**
>
> *All or at least some of these decisions would need to be relaxed to make a convincing paper.
>
> you are right, even if the focus of the paper is not on getting the best possible score on language modelling, different settings would make this point not only more convincing, but clearer.
>
> *The reasons for the use of the energy-based formulation are not clear to me. Is the energy-based model particularly well-suited to the random-projection setup, or are there other reasons for using it, independent of the use of random projections?
>
> This formulation is fundamental for the next step of the work in which we are removing the restrictions from the output layer and learning word probability distributions without prior knowledge of the vocabulary size. That said, the formulation is just the re-use of the embedding layer transposed. It removed an entire set of m x V parameters and got us better results in all our experiments so we decided to use it.
>
> * It looks like all the results are given on the test set. Did you not do any tuning on the validation data?
> Yes, all the parameters were tuned on the validation data. All the models were selected according to their validation data evaluation.
> The early stop criterion is also based on the validation data evaluation. We consider the model to converge when it cannot improve further on validation data. The models never saw the test set during training or tuning, otherwise we would be cheating and these scores would be irrelevant to compare different settings.

---

### Official Review · AnonReviewer3 · 2018-11-01
**confusing on the motivation and choices of experimenting models**

**Rating:** 4
**Confidence:** 3

**Review:**

This paper studied a random projection of word embeddings in neural language modeling. Instead of having |V| x m embeddings, the author(s) represented a word with a random, sparse, linear combination {1, 0, -1} of k vector of size m. The experiment on PTB dataset showed that k had to be somewhat close to |V| in order to achieve the comparable perplexity to a feed-forward NLM.

Overall, I am not sure what we could gain from this research direction. The advantage of this random encoding was to reduce the number of parameters for an embedding layer, but the results showed we gained much PPL from a 25% reduction in embedding size (Table 1). In addition, the fact that the random projections preserved the inner product (centered at zero) was probably not desirable. It might be more fruitful if these linear combinations were learned or sub-senses of words (e.g. [1]).

The experiments were quite extensive on the hyper-parameters and showed how the models performed under different settings. However, these were done using 1 dataset and also a simple feed-forward network (rather than LSTM). I can understand the point that training NNLM accelerates the experiments, but the author(s) should consider trying a simply LSTM model after the best settings had been discovered (e.g. Table 1). PTB also has a very unnatural vocabulary distribution as pointed out in [2]. Thus, it might be helpful to test the result on another dataset (e.g. WikiText).


Other comments
1. I do not get the point of bringing up NCE. Did you actually use NCE loss? Did you only refer to NCE as a weight tying which can be used in a standard XENT loss [3]? The first paragraph of 3.3 did not help clarify this point either.

2. In Figure 3, the baseline got different perplexity between 3(a) and 3(b).

3. Shouldn't random indexing produce non-uniform numbers of non-zero entries depending on alpha? Why did you have an exact number of non-zero entries, s, in the experiments?

3. Some typos
- "... is that instead of trying to probability ..." => "... tying ..."
- "... All models sare trained ..." => "... are ..."
- "... Tho get the feature ..." => ?

References
[1] S. Arora et al., 2016. Linear Algebraic Structure of Word Senses, with Applications to Polysemy
[2] S. Merity et al., 2016. Pointer Sentinel Mixture Models
[3] Y. Gal et al., 2015. A Theoretically Grounded Application of Dropout in Recurrent Neural Networks

---

> ### Author Response · Authors · 2018-11-27
> **some clarifications**
>
> * the fact that the random projections preserved the inner product (centered at zero) was probably not desirable. It might be more fruitful if these linear combinations were learned or sub-senses of words (e.g. [1]).
>
> Preserving the inner product means that the distribution of the features is not biased, if we keep adding words to the dictionary, the performance would degrade gracefully with the amount of compression. Perhaps a non-orthonormal basis would also work if the network compensates for the different distortions in the inner products. You are correct in assuming that other discrete building blocks could be more fruitful, but, we chose language modelling as a setting, not a task (see general response) as such, the building block chosen was the word. We could have chosen sub-words, or characters but the goal here is not the get the best possible language model but to understand a property of the mechanism.
>
> An interesting idea would be to actually use other information and encode it as random projections (e.g. syntactic dependency patterns). The amount of possible patterns is simply too large to be enumerated and as such the random projections would serve as unique "fingerprints" for unique "dependency patterns" that would be used as inputs.
>
>
> 1. I do not get the point of bringing up NCE...
>
> Approximations like NCE (in conjunction with random projections) would allow us to remove the restriction in the output layer. We want to imply that our proposal is not incompatible with NCE, but we did not yet explore it so, to make the paper more self-contained it is probably best to leave this out.
>
> 2. In Figure 3, the baseline got different perplexity between 3(a) and 3(b).
>
> we were trying to cram different experiments (with different regularization) in the same figure which is understandably confusing and needs to be corrected.
>
> 3. Shouldn't random indexing produce non-uniform numbers of non-zero entries depending on alpha? Why did you have an exact number of non-zero entries, s, in the experiments?
>
> yes but not necessarily. Alpha can be used to control the expected proportion of non-zero entries, but as long as the probability of a sparse configuration is random uniform, our mechanism guarantees that any sampled index is almost orthogonal to any other sampled index, so it's easier achieve the same while guaranteeing the sparsity in the inputs.

---

### Official Review · AnonReviewer1 · 2018-11-02
**Experiments and novelty need improvement**

**Rating:** 3
**Confidence:** 4

**Review:**

This paper presents some experiments using random projections instead of embeddings from a 1-of-V encoding.  Experiments on the Penn TreeBank benchmark data set show that in a feed-forward language modeling architecture similar to that of (Bengio, 2003), the random projections substantially reduce the number of parameters of the model while not harming perplexity too much.

The paper would need to be improved substantially in order to appear at a conference like ICLR.  First, the novelty of the approach is limited -- the approach amounts to using a sparse integer layer instead of a floating-point layer within a feed-forward architecture.

Second, and more importantly, the experiments need to be re-done to better measure the practical impact of the techniques.  First, larger data sets such as Wikitext-2 and Wikitext-103, and/or the billion-word benchmark, are needed to understand how well the approach works in practical LM settings.  Second, the paper needs to use more state-of-the-art architectures.  Language modeling is a fast-moving field, so the very latest and greatest techniques are not strictly necessary for this paper, but at least midsize LSTM models that get scores in the ~80 ppl range for Penn Treebank are important, otherwise it becomes very questionable whether the results will provide any practical impact in today's best models.  Finally, the paper needs to compare its parameter-reduction approaches against other compression and hyperparameter optimization techniques.  Changing the number/sizes of the network layers or using sparse weight matrices (perhaps with sparsity-inducing regularization) would be natural ways to reduce the parameter space.

In my opinion, due to how many researchers are and have been looking into improvements of language modeling, the authors may find it hard to break new ground in this direction.

Minor
In the start of Section 3, it is not clear why having the projection be sparse is desired.  Later, space (and time) efficiency is revealed as the motivation for the sparsity, but it would be helpful if the paper said this earlier.
Equation 6 seems to have an error, the probability should be P(w_t | w_t-1...) instead of P(w_t , w_t-1...) if this is to represent the standard LM objective (the probability of the corpus).
Sec 3.3: "all models sare"

---

> ### Author Response · Authors · 2018-11-27
> **response**
>
> * models that get scores in the ~80 ppl range for Penn Treebank are important.
>
> we agree with the advice but not with the justification. We explain why in the general response: our goal is not to get good language models, but to use language modelling as a setting to test a property of a mechanism that is proposed. The perplexity becomes a way to observer the effect of a mechanism and not the goal itself. Moreover, (not in this case but) the architectures used to achieve better scores on given datasets are so over-parametrized that it's hardly reasonable to assume that the improvement justifies the cost of accommodating huge models overfitted to a particular dataset (and sometimes to a particular dataset configuration)
>
> That said, we agree that using different architectures would strengthen our point and make the paper more convincing. Also, using different datasets would help us demonstrate that the effect of the proposed mechanism is data-independent. We are also considering it's application to a different set of tasks in the future.
>
> We did follow reviewer recommendations and performed experiments with LSTMs and QRNN (slightly faster) along with WikiText (which is larger but not intractable), unfortunately we couldn't accommodate all the analysis and changes in time.
>
> * its parameter-reduction approaches against other compression and hyperparameter optimization techniques.
>
> We recognize that the focus on parameter reduction was perhaps counter productive to making the goal or this work clear. It is a byproduct of the technique, but modelling discrete distributions without prior knowledge of how many classes one might encounter is the main issue we are trying to solve. We could do that by using character-level or sub-word tokens, but again, the goal is not --solely-- language modelling as a task. The mechanism is applicable to settings where the number of possible input patterns is too large to instantiate as a parameter table (embeddings), but where the number of patterns that actually occur could actually more "reasonable". Meaning that as long as the "world" is not random uniform, we can make predictions.

---

> > ### Comment · AnonReviewer1 · 2018-12-05
> > **Thank you for the response**
> >
> > Given that the goal isn't language modeling I think the paper might be better served if you targeted a different domain with large numbers of objects---one that is less well-studied than language modeling and where object structure like subwords isn't available.  Unfortunately, I don't have a concrete suggestion for such a setting (collaborative filtering data sets have large numbers of relatively atomic objects, but that task is well-studied).  Another alternative would be to set aside trying to demonstrate practical impact and instead focus more on synthetic data; although of course as a reviewer I would then expect to see expanded novelty and more theoretical results.  I appreciate you taking the time to provide a response, best of luck.

---

> > > ### Author Response · Authors · 2018-12-05
> > > **thank you again**
> > >
> > > What we mean is that while language modelling is the setting, the absolute best perplexity scores are secondary to our exploration. It's tricky to convey this idea because the narrative is that progress is measured in terms of SOTA scores, and this is (sometimes) counter-productive to learning anything about a specific problem. In the extreme, it is not uncommon to find claims about the importance of improving by 1 point of perplexity, which is ... perplexing.
> > >
> > > * "one that is less well-studied than language modeling and where object structure like subwords isn't available"
> > >
> > > We strongly disagree with the view of language modelling purely as a task to be "solved". It is precisely because of how general the setting is, that it is perfect to test our assumptions about sequence modelling. Language modelling provides after all, a sequence modelling setting where the sequences follow complex patterns. Moreover, while the task is rather "useless" on its own, it reveals many techniques that can be used to solve a wide array of practical tasks (some beyond language modelling)
> > >
> > > The fact that the problem is well studied allows us to create reasonable baseline models, to know what to expect in a typical scenario where no information compression techniques are used. That said, we still agree that we must explore other common architectures to highlight the relationship between model performance and the inner product distortion of the random projections.
> > >
> > > again, thank you for your response

---

### Author Response · Authors · 2018-11-27
**General Response**

We would like to thank all the reviewers for their valuable comments on this
submission. While we disagree with some points that were made (which we would
like to address individually) overall, your reviews helped us understand the
reception this kind of work would have and which points should be make
absolutely clear.

Notwithstanding some issues like typos, and notation that needs to be corrected,
we identified 3 fundamental points that must be improved or changed in the
presented work:

1. we should use other (more "state-of-the-art" architectures) e.g. LSTM
2. we should extend this exploration to other (larger/different) datasets
3. there seems to be a misunderstanding (to our fault) about the goal of this
study.

In this work, language modelling is not a goal, but a setting under which
we test some assumption about the property of the mechanism we propose. What we
mean by this is that the perplexity scores we present are used as grounding
(difference between baseline and modified model) rather than goal. Moreover
random projections would never allow us to get models with better perplexity
because it acts as a form of compression.

We used a simple architecture and dataset to promote reproducibility (code would
also be made available) but the main point is that the mechanism we propose is
actually data and architecture-independent.

We didn't mention this in the paper explicitly, because we thought it should be self-contained,
but this work is part of a larger goal to make inference possible without prior knowledge
of how many discrete symbols one might find when modelling discrete probability distributions.
This also means that the mechanism is not intended to be applicable to language modelling only,
but the setting is highly complex and has a wide array of related tasks which makes it interesting
as a starting point.

There are two things that don't allow us to achieve this:

1. The neural network architecture input layers depend on a vocabulary size
2. The output (e.g. softmax and cross entropy loss) also depend on a vocabulary size.

In this work we intended to explore a solution to 1. Obviously we could use character-based or
sub-word based modelling. But the goal is to work with a scenario where such reductions might
not be possible. In any case, the random indexing principle could be used for any discrete building block.

We found that learning can be done without using a single unique vector representation for each unique
discrete input. We also found that the perplexity decays exponentially with the size of the vector
representations used (embeddings). This was the main point, it follows from the properties of random
projections, but how this mechanism would influence the performance of a neural network model was
not explored in the literature (to our knowledge)

Since we want to show this property to be architecture independent, we concede that perhaps using an
additional architecture would help us to make this point more clear.

The energy-based formulation of our proposal is actually an integral part of the
second problem (substituting the output layer with an approximation that does
not require the vocabulary to be known a prior (like NCE). Nevertheless, the energy-based
formulation, is simply the sharing of the embedding weights transposed to compute
the model logit (and yielded better results with less parameters).

Finally, we were able to perform experiments with new architectures and
additional datasets (unfortunately not in time for the re-submission).
More specifically, we tested LSTM networks and QRNN networks (with similar
performances to LSTMs but much faster), we also used Bayesian Optimization to
solve some problems we had with hyperparameter tuning. As for datasets, we used
WikiTexT-102 and wikiText103 (pending experiments): the first being 2 times larger than PTB,
the second 100 times larger.

Again, thank you for your comments, we hope this clears up some issues raised in
the reviews. Also feel free to raise any additional issues or questions
regarding this work, we greatly appreciate the feedback/discussion.

---

### Meta-Review · Area_Chair1 · 2018-12-13
**authors must make it as easy as possible for readers to understand the contribution**

**Confidence:** 5
**Recommendation:** Reject

**Metareview:**

There is a clear reviewer consensus to reject this paper so I am also recommending rejecting it. The paper is about an interesting and underused technique. However, ultimately the issue here is that the paper does not do a good enough job of explaining the contribution. I hope the reviews have given the authors some ideas on how to frame and sell this work better in the future.

For instance, from my own reading of the abstract, I do not understand what this paper is trying to do and why it is valuable. Phrases such as "we exploit the sparsity" do not tell me why the paper is important to read or what it accomplishes, only how it accomplishes the seemingly elided contribution. I am forced to make assumptions that might not be correct about the goals and motivation. It is certainly true that the implicit one-hot representation of words most common in neural language models is not the only possibility and that random sparse vectors for words will also work reasonably well. I have even tried techniques like this myself, personally, in language modeling experiments and I believe others have as well, although I do not have a nice reference close to hand (some of the various Mikolov models use random hashing of n-grams and I believe related ideas are common in the maxent LM literature and elsewhere). So when the abstract says things like "We show that guaranteeing approximately equidistant vector representations for unique discrete inputs is enough to provide the neural network model with enough information to learn" my immediate reaction is to ask why this would be surprising or why it would matter. Based on the reviews, I believe these sorts of issues affect other parts of the manuscript as well. There needs to be a sharper argument that either presents a problem and its solution or presents a scientific question and its answer. In the first case, the problem should be well motivated and in the second case the question should not yet have been adequately answered by previous work and should be non-obvious. I should not have to read beyond the abstract to understand the accomplishments of this work.

Moving to the conclusion and future work section, I can see the appeal of the future work in the second paragraph, but this work has not been done. The first paragraph is about how it is possible to use random projections to represent words, which is not something I think most researchers would question. Missing is a clear demonstration of the potential advantages of doing so.